# Phytochromes facilitate social behaviour in marine diatoms

Joan S. Font-Muñoz [1,2] ✉, Marianne Jaubert [3], Marc Sourisseau [2], Idan Tuval [1], Benjamin Bailleul[3,4], Carole Duchêne [3,5], Gotzon Basterretxea[1] & Angela Falciatore[3]

The phytochrome superfamily comprises photosensory proteins that enable organisms to perceive changes in light intensity and quality and is widespread across plants, fungi, algae, and microbes. In terrestrial plants, phytochromes sense red and far-red light to regulate key developmental and physiological processes. In marine environments, however, where red and far-red wavelengths penetrate only the upper few meters of water, the function of phytochromes has remained unclear. Recent work shows that diatom phytochromes exhibit photoreversible responses across a broad spectral range, extending beyond red and far-red, suggesting a role in underwater light sensing. Here, we examine the role of phytochromes in light perception and collective behavior in the marine diatom *Phaeodactylum tricornutum*. Comparing wild-type and phytochrome knockout strains under different light wavelengths reveals that activation of phytochromes by blue or far-red light synchronizes cell movements into a coordinated "wobbling dance." This behavior is absent in phytochrome-deficient mutants, demonstrating the essential role of phytochromes. Our results further suggest that this collective motion involves intercellular communication, potentially mediated by variable red and far-red autofluorescence. Together, these findings uncover a previously unrecognized light-driven social behavior in marine diatoms and highlight the ecological significance of phytochrome-mediated communication in microbial communities.

Photosynthesis is an essential biological process that sustains life on Earth by enabling autotrophic organisms to transform solar energy into chemical energy, which in turn serves as a vital source of sustenance for other living organisms. The effectiveness of photosynthesis relies, among other factors, on the ability of autotrophs to adapt to fluctuating environmental sunlight conditions. To this end, they have evolved highly specialized sensory molecules that allow the perception of incoming environmental signals and transduction into biochemical outputs that shape organism responses[1,2].

Phytochromes are a group of photoreceptor chromoproteins that enable plants and some photosynthetic and non-photosynthetic microbes to detect and adapt to changes in light intensity and quality[3]. In terrestrial plants, phytochromes (PHYs) are capable of sensing light in the red ($R = 660$ nm) and far-red ($FR = 780$ nm) regions

[1]Department of Marine Ecology, Instituto Mediterráneo de Estudios Avanzados, IMEDEA (UIB-CSIC), Miquel Marqués 21, 07190 Esporles, Illes Balears, Spain. [2]Dynamic of Coastal Ecosystems, DYNECO, Ifremer, Technopôle Brest Iroise, Plouzané, France. [3]CNRS, Sorbonne Université, Institut de Biologie Physico-Chimique, Laboratoire de Biologie du chloroplaste et perception de la lumière chez les microalgues, UMR7141, Paris, France. [4]UMR7144 AD2M, ECOMAP, Station Biologique de Roscoff, Sorbonne Université/CNRS, Roscoff, France. [5]Present address: Department of Algal Development and Evolution, Max Planck Institute for Biology Tübingen, 72076, Tübingen, Germany. ✉e-mail: jfont@imedea.uib-csic.es

of the spectrum[4]. PHYs provide essential plasticity to plant survival by activating specific signaling cascades and enabling the active adjustment of vital processes such as germination, growth, development, phototropism, and associated metabolic pathways[5,6]. Plant PHYs are also capable of perceiving changes in the *R/FR* ratio of ambient light caused by neighboring vegetation, initiating shade avoidance and acclimation reactions[7]. In some human pathogenic bacteria, PHYs-mediated light signals also integrate quorum-sensing responses to control collective behaviors and diverse lifestyles[8].

PHYs have also been recently found in a broad range of aquatic microorganisms. Their discovery first came as a surprise because *R* and *FR* wavebands derived from sunlight radiation are scarce in aquatic environments, due to their rapid attenuation by water absorption[9,10], opening questions about the mechanisms of action and function of these photoreceptors[11]. The more recent characterization of phytochromes from diverse eukaryotic algae provided some answers to this conundrum. Extensive spectral tuning and blue-shifted absorption spectra have been described in PHYs of some prasinophyte and glaucophyte species, leading to the hypothesis of an adaptation of the photoreceptor absorption properties to the blue-rich aquatic environment[12]. On the contrary, conserved *R* and *FR* light absorption spectra have been described for the phytochrome of diatoms[13], one of the most prominent and diversified phytoplankton groups in the ocean, which are distributed worldwide and play a key role in global biogeochemical cycles[14,15]. Unforeseen roles for *R* and *FR* PHYs-mediated detection have been hypothesized, such as the perception of fluorescent light emitted by nearby photosynthetic congeners, potentially providing valuable information on cell densities, for example, during algal blooms[13,16,17], and stimulating various aspects of their cellular metabolism, such as iron uptake[17]. Recent studies, however, indicate that the Diatom Phytochromes (DPH), despite the conserved *R* and *FR* light absorption[13], can also exhibit photoreversible responses across the whole available underwater light spectra, suggesting an additional potential role of DPH as an optical depth sensor[18] but not ruling out other functional roles.

Diatom lifestyle diversity and survival success in contemporary oceans are attributed to their exceptional ability to adapt to highly dynamic aquatic environments using efficient mechanisms to respond to environmental changes[19-22]. For example, several diatom species can cope with highly variable light conditions, suggesting that diatoms are capable of perceiving, responding, and likely, anticipating light variations and that they possess suited underlying molecular systems

mediating light responses[23,24]. Recent studies have also shown that several pennate marine diatoms are indeed capable of perceiving and using external light signals to synchronize intracellular processes driving their unsteady sinking behavior[25,26]. However, the molecular mechanisms underpinning these photoresponses and the role of PHYs in light-mediated behavior at the population level are not yet established.

Here, we use several independent *P. tricornutum* dph-knockout lines, together with laser diffractometry, to evaluate the role of photosensing in the light-mediated development of collective cell behavior[27-30]. We experimentally demonstrate that photoreception by DPH is required to establish a coordinated social pattern in populations of *P. tricornutum* suspended in the water column. The possibility that microbial organisms use endogenous light signals to perceive and communicate with neighboring homologous cells for the development and propagation of coordinated behavior is also discussed.

## Results and discussion

Pelagic diatoms with elongated shapes characteristically display coherent vertical cell orientation and synchronized wobbling when sinking under continuous illumination[25,26,31]. In the case of nearly elongated cells, this behavior can be characterized by laser diffractometry (LD) using the relative variation of two size bands associated with the major and minor cell axis lengths[30,31]. Due to the 2D projection of the LD measurements, the *Ratio* between these signals is a scalar proxy for the mean orientation of the cell population (see M&M). As shown in Fig. 1 (see also Fig. S1; Supplementary Data 1, 5), in the diatom *P. tricornutum*, the *Ratio* time series displays a pronounced wavelength dependence of synchronized wobbling, with strong responses in the *B* and *FR* regions, but minimal responses elsewhere in the light spectrum. This spectral dependence is consistent with that described by Duchêne et al.[18] for DPH-mediated gene expression (Fig. 1B). No coherent wobbling is observed in dph-knockout strains (*5D1* and *4F1*), while their respective control lines (*Tc5A1 and Tc4E1*, transformed but non-mutated in DPH[13]) exhibit a synchronized collective response under *B* light, which is indistinguishable from that observed in the wild-type (*WT Pt1*; Fig. S2; Supplementary Data 6). These results strongly support that DPH photoperception is strictly required for light-induced synchronized cell wobbling.

A key feature of PHY-mediated control is its ability to scale the magnitude of a response according to the wavelength ratio. Since, in *P. tricornutum,* DPH was shown to be activated by *B* or *FR* and inactivated

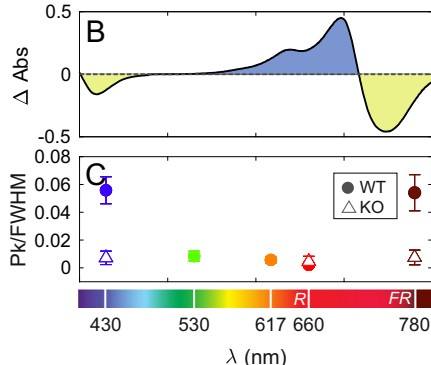

**Fig. 1 | Experiments with continuous exposure to light. A** Power spectrum of *P. tricornutum* (WT) *Ratio* obtained with laser diffractometry experiments (see Fig. S1 for the original time series). The dotted line represents the measured data, and the solid line corresponds to the Gaussian fit, as detailed in the Supplementary Materials. Colors indicate the wavelength with which cells were illuminated (430, 530, 617, 660, and 780 nm). Light irradiance was kept constant for all wavelengths (8 μmol photons m$^{-2}$ s$^{-1}$). **B** DPH deactivation/activation (green/blue areas) spectrum (Δ Abs) for the WT resulting from absorption differences between the

synthesized active state of DPH and the inactive conformation (adapted from Duchêne et al.[18]). **C** Peak amplitude (*Pk*) of the power spectrum (shown in A) normalized by the full width at half maximum (*FWHM*) of the power spectrum for the wavelength of the light used (arbitrary units). Solid circles correspond to experiments with wild-type (WT) and transformed, but DPH non-mutated strains, and empty triangles correspond to experiments carried out with DPH knockout strains (KO). Data are shown as mean ± SD (*n* = 3 independent biological replicates).

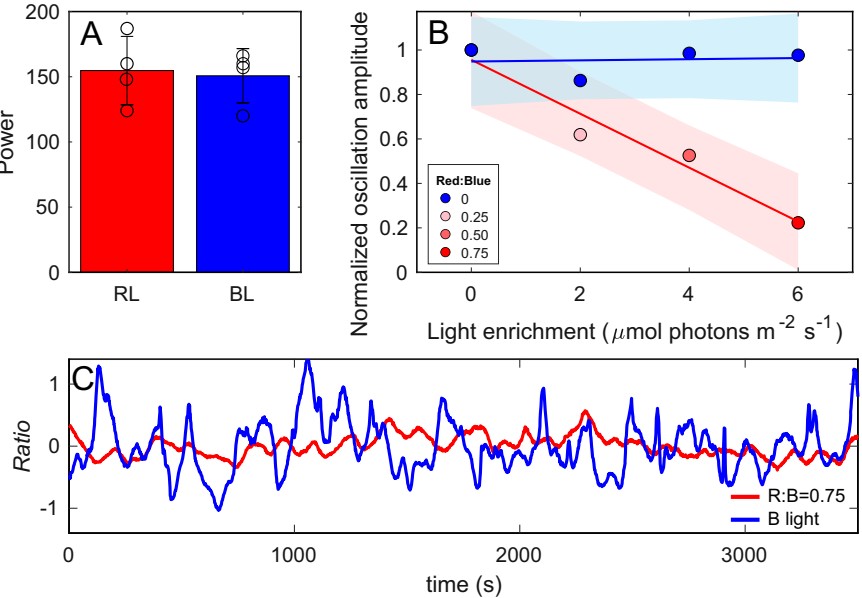

**Fig. 2 | Diatom response to continuous exposure to blue and red light. A** Mean period of oscillation of WT cells for experiments with different ratios of *R:B* light and the *B* light controls, bars show mean ± SD (*n* = 3 independent biological replicates). **B** Normalized oscillation amplitude for experiments with enhanced light (under *B* light background of 8 μmol photons m⁻² s⁻¹) at different ratios of *R:B* light and the controls with *B* light. Red and blue lines show the linear fits to the experimental data; shaded bands indicate the corresponding 95% confidence intervals. **C** Normalized *Ratio* time series for experiments with a 0.75 *R:B* ratio and *B light* control of the same intensity.

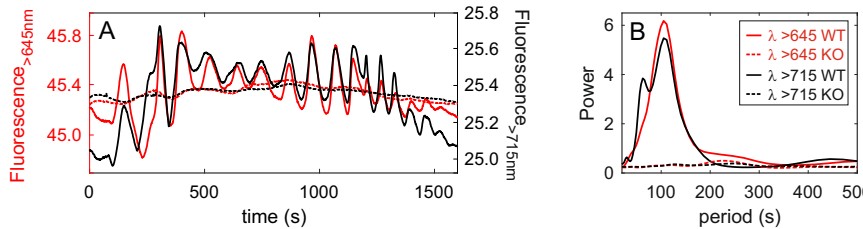

**Fig. 3 | Natural fluorescence emission. A** Time evolution of the fluorescence (x10⁻³ μmol photons m⁻² s⁻¹) emitted at two different spectral bands (λ > 645 nm, red line, and λ > 715 nm, black line) in a suspension of *P. tricornutum* (*WT* solid lines, DPH knockout dashed lines) illuminated with 430 nm light (30 μmol photons m⁻² s⁻¹) and **B** the corresponding power spectrum.

by *R* light[18], we tested whether the described collective behavior is hindered by changing the relative intensity of *B* and *R* light. Starting with the intensity of *B* light used in Fig. 1, we simulated the effect of decreasing depth in the water column by increasing *R* light (shallower waters), and observed inhibition of the synchronized wobbling amplitude (Fig. 2, see also Fig. S3; Supplementary Data 2, 7) while wobbling frequency was unaffected (Fig. 2A). A similar response was observed in experiments performed with varying *FR* to *R* ratios (Fig. S4; Supplementary Data 8). To confirm that this was a response to *R* light and not just the consequence of modified light quantity, we also performed control experiments with increased *B* (or *FR*) light intensity and observed no change in the response (Figs. 2B, S4). Our results suggest that, similarly to the proposed in situ modulation of DPH activation by optical depth throughout the water column[18], synchronized wobbling is also modulated by blue-to-red ratio. As a consequence, it would be favored at depth, an environment dominated by *B* light.

While the triggering mechanisms of the described cell behavior rely on ambient *B* light availability and detection by DPH, the mechanism through which independent cells freely suspended in the water column coordinate their movements remains challenging. Font-Muñoz et al.[25] framed the question within the context of the theory of weakly coupled oscillators for phase locking[32], which requires a

harmonic coupling signal compatible with the cell wobbling frequency (ω; Fig. 3; Supplementary Data 3). It can be speculated that either hydrodynamic physical disturbances in the flow field surrounding individual cells or the modulation of unknown dissolved infochemicals govern these cell-to-cell interactions[33–35]. However, these are unlikely to generate coherent forcing at the involved spatial and temporal scales. Alternatively, *B* light-excited cell autofluorescence emitted in the *R* and *FR* spectral regions provides a plausible explanation for intercellular communication, inducing the observed synchronized movements. Autofluorescence emission by photosystem II is a by-product of photosynthesis that is cost-efficient and almost instantaneous, and depending on cell morphology, can provide a cell density-dependent but directional signal of the correct frequency[16,17]. Although in the sea the intensity of fluorescence emission is always very small compared to the ambient light, it could still convey precious information and be perceivable locally by neighboring cells as a weak coupling signal above the *R* light background. Indeed, the measured emitted chlorophyll fluorescence signal in each arbitrary direction is modulated by the cell's relative orientation, thus exhibiting a time modulation at the exact frequency of the cell's wobbling oscillation, as displayed in Fig. 3. This periodic light signal comprises both *R* and *FR* spectral components, whose temporal signals are highly similar (Fig. 3B). It is worth noting that while dph-knockout strains

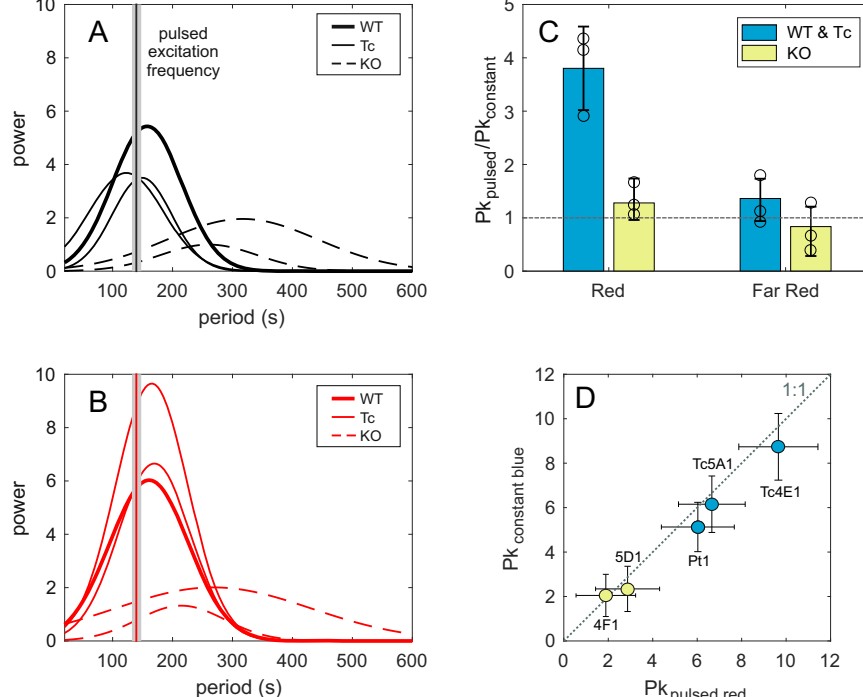

**Fig. 4 | Pulsed light experiments.** Power spectrum of the *Ratio* for experiments that mimic natural fluorescence emission. The thick solid lines correspond to the *WT* strains, the thin solid lines correspond to the *Tc* strains, and the dashed lines correspond to the DPH knockout strains. The gray band indicates the period of light oscillation. Panel **A** shows experiments with far-red light (780 nm), and panel **B** shows experiments with red light (660 nm). **C** Differences in collective behavior when using pulsed light versus continuous light: comparison of the peak amplitude of the power spectrum of *Ratio* under pulsed light (red or far-red, -140 s) to the peak amplitude under constant light (red or far-red), values close to 1 indicate similar collective behavior, bars represent mean ± SD (*n* = 3 independent biological replicates). **D** Peak amplitude of the power spectrum of experiments with constant exposure to *B* light compared to pulsed red light (-140 s). The abbreviations correspond to the different strains, and data are shown as mean ± SD (*n* = 3 independent biological replicates).

autofluoresce with the same spectral qualities as *WT* -at room temperature-, they do not exhibit a population-level coherent fluorescence signal modulation. It should also be stressed that illuminating an algal suspension with an exogenous monochromatic light does not guarantee a homologous light environment since both Raman scattering and, more intensely, chlorophyll autofluorescence cause the emission of photons shifted in frequency relative to the incident light, thereby modifying the spectral composition of the surrounding light field[16,36]. Exogenous light and chlorophyll-emitted light signals cannot easily be decoupled. This opens the question of whether DPH mediates cellular responses only to exogenous ambient light or whether they are also involved in the perception of time-modulated autofluorescence.

The idea that PHYs enable algae to sense neighboring cells through the perception of cell autofluorescence is not novel[13,16] but, to the best of our knowledge, has yet to be confirmed. We set out to test this hypothesis by interrogating cells only with pulsed *R* and *FR* light that mimics the measured fluorescence signal naturally emitted by *P. tricornutum* cells (see Fig. 3A). In Fig. 4, we show a striking contrast in the response of the *WT* and the dph-knockout strains to the same pulsed stimulus, both under *R* and *FR* illumination. While *WT* and *Tc* cells are readily entrained by pulsed light, the wobbling of dph-knockout strains remains largely incoherent (Fig. 4A, B; Supplementary Data 4). These results indicate that DPHs are not only essential for triggering collective behavior under constant light illumination, but they are also key to the synchronization process, potentially implying their involvement in the ability of cells to perceive the light emission of their neighboring congeners and thereby contribute to the observed collective behavior by propagating the signals in the population. Full characterization of this function and its ecological relevance remains to be elucidated.

Finally, it is important to note that, while a collective photoresponse is not elicited by illumination under constant *R* light in our experiments (see Fig. 1C), a periodic modulation of the signal with frequencies around natural wobbling frequencies (-140 s) suffices to induce strongly coherent cell responses (Fig. 4C). In fact, these are as coherent as those observed under constant *B* (or *FR*) illumination (Fig. 4D). Notably, DPH-knockout strains do not exhibit collective responses in either case. This differential photoresponse, not observed at any other wavelength, hints at a possible unknown DPH signal transduction pathway and PHY-facilitated mechanism for light perception under quickly fluctuating light environments.

We have found that phytochromes are involved in light-driven coordinated diatom wobbling, a collective behavior modulated by the effective wavebands activating DPH[18], and therefore enhanced in *B* light-enriched environments. Although the function of this collective dynamics in the dilute marine environment is still unclear, we can speculate that it could influence key ecological processes such as sedimentation rates, light harvesting, or increased cell-cell contacts, thereby favoring sexual reproduction in pelagic environments[25,26], especially in the deep, non-turbulent layers of the water column[31,37,38].

The analysis presented here immediately suggests a number of specific experimental investigations. Chief among them are detailed studies of DPH involvement in the photoresponse to pulsed *R* light. This can be tested with experiments at the single-cell level in which photoresponses to exogenous light can be readily distinguished from signaling from neighboring cells. Open issues include the origins of this regulation, the comprehensive characterization of the perceived stimuli, along with the possible interplay of these processes with other biotic and abiotic signals, as those observed in the quorum sensing of bacteria due to high cell densities[39]. Additionally, our results under

pulsed *R* and *FR* light suggest that oscillations of autofluorescence by cell wobbling could be involved in the propagation of the DPH-mediated response. Nonetheless, earlier observations of collective oscillations in pennate diatom species that are not known to express DPHs[25], suggest that DPH may function in coordination with other, yet undiscovered photoreceptors. Overall, our findings provide strong evidence of the relevance of light signaling in shaping microorganism interactions and social behavior, and, at a higher hierarchical level, as a structuring element of life in the ocean.

## Methods

### Cell cultures

The wild-type *Phaeodactylum tricornutum* cells and transgenic lines were grown in f/2 medium[40] at 18 °C under a 12-h-light/12-h-dark cycle and 80 μmol photons $m^{-2} s^{-1}$, in algal incubators. The cultures were transferred once a week (dilution factor x8) to keep the cells healthy and in their exponential growth phase at maximum concentrations. All the experiments were done at a concentration of $3 \times 10^8$ cells/L, with cells in the exponential phase of growth; each experiment was carried out in triplicate. DPH knockout lines were obtained by TALEN-mediated genome editing as described in Fortunato et al.[13]. Their respective DPH non-mutated lines (derived from the same cell transformed with TALEN vectors but not having undergone a mutagenic event on the DPH gene) were used as a control for DPH-specific phenotype[13].

### Experimental set-up

Experiments were carried out in an in-house built set-up for the concomitant measurement of particle size distribution, orientation, and chlorophyll autofluorescence as in Font-Muñoz et al. (2021)[25]. The small volume ~100 ml flow-through chamber of a LISST-100x laser diffractometer (Sequoia Scientific) was adapted to include an additional light-line source. This light was modulated both spectrally, using a monochromator, and temporally, and aligned orthogonally to the laser diffraction axis. Two Thorlabs PM16-130 power meters were integrated into the setup to monitor this secondary illumination. The modified system enables the investigation of a broad spectrum of experimental conditions, providing simultaneous sub-second resolution measurements of both cell orientation and light emission[25].

Before each experiment, the LISST-100X chamber was pre-cleaned and filled with the desired v/v concentration of cell culture. Agitation was induced by a 2 cm long magnetic bar powered by the built-in speed controller of the chamber at minimum speed. Experiments consisted of an initial mixing phase where agitation was turned on (100 s), followed by a sedimentation phase where the system could evolve without any external disturbance.

**Diatom response to light.** To study the effect of light conditions on cell orientations, the cells and collective behavior were illuminated with continuous light of different wavelengths (430, 530, 617, 660 and 780 nm, mod. M430L5, M530L4, M617L5, M660L4 and M780L3 from Thorlabs) with an intensity of 8 μmol photons $m^{-2} s^{-1}$ for 2 h.

**Diatom response to composed blue and red light.** To study the responses of cells to composed continuous red and blue light, cells were illuminated with 430 nm at 8 μmol photons $m^{-2} s^{-1}$ and 660 nm at different intensities (2, 4, and 6 μmol photons $m^{-2} s^{-1}$). We performed experiments using monochromatic blue light at the same total intensities (10, 12 and 14 μmol photons $m^{-2} s^{-1}$) as a control. We used a long-pass dichroic mirror 550 nm cut-on (DMLP550 from Thorlabs) to combine the two light sources.

**Diatom response to pulsed red and far-red light.** To study the response of *P. tricornutum* to pulsed red light, cells were illuminated by

a 660 nm and 780 nm LED (M660L4-C and M780L3, Thorlabs) through a neutral density filter (ND10A Thorlabs) used to attenuate light intensity down to a maximum value of ~8 μmol photons $m^{-2} s^{-1}$. This intensity was then modulated sinusoidally with periods of cell wobbling natural frequency (~140 s) by coupling the controller triggering the excitation LED source to a pulse generator.

**Fluorescence signal variations measurements.** Variations in the cells' chlorophyll autofluorescence were measured by illuminating the culture with an excitation blue light (430 nm at 30 μmol photons $m^{-2} s^{-1}$) for 45 min while the cells settled without disturbance. The emitted *R* and *FR* light coming from the cells was measured every second by the orthogonal power-meters (PM16-130 from Thorlabs) via high-pass *R* and *FR* filters (FGL645M, Thorlabs; $\lambda > 645$ nm and FGL715M, Thorlabs; $\lambda > 715$ nm).

### Laser diffractometry

Laser diffractometry (LD) measurements were used to obtain the particle volume concentration by size ranges (i.e., volume of particles in the seawater per unit volume of seawater) using a technique based on laser diffraction theory. The LISST-100X employed in this study uses a 670 nm collimated laser beam to illuminate the suspended particles, and a 32-ring detector measures the intensity of the scattered light, corresponding to 32 different size classes logarithmically spaced. The angular pattern of optical scattering depends on the physical characteristics of the particles (e.g., size, shape and orientation), and is used to calculate the particle volume concentration in these size classes[41]. LD can be used to adequately characterize the different dimensions of non-spherical particles in specific orientations[28,29,31]. Using this property, a method to infer the preferential orientation of particles in a suspension has been recently described[30]. Hereby, we use this method to characterize the preferential orientation of diatoms under distinct lighting conditions: LD measurements of nearly spheroidal cells are interpreted in terms of the relative variation of two size bands representing the known major and minor cell axes ($r_1$ and $r_2$) of *P. tricornutum* (major axis *VD2* = 7.33–14 μm and minor axis *VD1* = 2.5–3.5 μm). As cells orient in different directions, the signal in the size bands representing these axes presents an opposite behavior. Hence, we can confidently compute the ratio between the signal from each of these two size bands, $Ratio(t) = (\Sigma VD_1)/(\Sigma VD_2)$, as a scalar proxy for cell orientation[25,30,31].

### Determination of spectral peak

We have used a standard and robust criterion to discriminate and characterize the main component of each power spectrum calculated from the Fast Fourier Transform (FFT) of the time-series of the *Ratio*. The criterion is based on conventional signal processing tools designed to locate peaks and arrange them by their relative prominence (*Pro*) in the signal (*Pro* = 6.2 ± 2.12; 2.4 ± 0.95; 2.38 ± 0.88; 0.73 ± 0.26; 5.12 ± 2.23, for the different wavelengths of the experiment, respectively). For signals with a variance of the relative peak prominence above one, std (*Pro*) > 1 (i.e., when a main peak can be isolated), we stick to a Gaussian fit to the main peak. For std (*Pro*) < 1 (i.e., when no peak stands out in a statistically significant way above the others), a single Gaussian is fitted across the full spectrum. Although a refined analysis of the secondary peaks might be of eventual interest, we do not consider that it has an impact on the results presented in this manuscript, so we defer this to future work.

## Data availability

Source data are available at the Zenodo repository (https://doi.org/10.5281/zenodo.18282507). All other relevant data are in the Supplementary Information and Supplementary Data files. Source data are provided with this paper.

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

## Acknowledgments

This research was supported by the ISblue project, Interdisciplinary graduate school for the Blue Planet (ANR-17-EURE-0015), and co-funded by a grant from the French government under the program "Investissements d'Avenir" and a grant from the Regional Council of Brittany (SAD program). J.S.F-M was supported by funding from "Margalida Comas" (PD/018/2020) and "Vicens Mut" (PD/055/2023) postdoctoral fellowships from Govern de les Illes Balears and Fondo Social Europeo. Funding from PID2022-143018NB-I00 and PID2019-104232GB-I00 grants from the Spanish Ministerio de Ciencia e Innovación (MICINN), the Agencia Estatal de Investigación (AEI), grant RGP007/2024 from the International Human Frontier Science Program Organization with the award (https://doi.org/10.52044/HFSP.RGP0072024.pc.gr.194152), and the H2020-MSCA-ITN-2020 PHYMOT is acknowledged. The present research was carried out within the framework of the activities of the Spanish Government through the "María de Maeztu Center of Excellence" accreditation to IMEDEA (CSIC-UIB) (CEX2021-001198-M). MJ acknowledges the CNRS MITI interdisciplinary program 'Lumiere et Vie'

and '80PRIME'. BB acknowledges support from the European Research Council (ERC) PhotoPHYTOMIX project (grant agreement No. 715579) and ERC Proof-of-Concept PALMADS (grant agreement No. 101158298). AF was supported by funding from the Fondation Bettencourt-Schueller (Coups d'élan pour la recherche française-2018), the "Initiative d'Excellence" program (Grant "DYNAMO," ANR-11-LABX-0011-01). J.S.F-M expresses his heartfelt gratitude to Susanna Castilla for her constant support and encouragement throughout the development of this research.

## Author contributions

J.S.F-M. conceptualized the study following discussions with A.F., M.J., M.S., and C.D. A.F. and M.J. contributed to the conceptual framework. J.S.F-M., A.F., M.J., and I.T. designed the experiments, with C.D. providing the mutant strains. J.S.F-M. performed the experiments and analyzed the data. J.S.F-M. and I.T. interpreted the results and designed the figures. B.B. proposed additional experiments, which were performed and integrated into the study by J.S.F-M. J.S.F-M. generated the figures and wrote the original draft. G.B. contributed to the interpretation of the results and edited and refined the figures. All authors J.S.F-M., A.F., M.J., I.T., B.B., C.D., M.S., and G.B. contributed to manuscript revision and discussion.

## Competing interests

The authors declare no competing interests.
