## [Transparent Peer Review file · Nature Communications]

Phytochromes facilitate social behaviour in marine diatoms

Corresponding Author: Dr Joan Salvador Font-Muñoz

Version 1:

Reviewer comments:

Reviewer #1

(Remarks to the Author)

Manuscript#: NCOMMS-24-74582A-Z

The authors tested the cell response of *Pheodactylum tricornutum* as a representative for pelagic diatoms with an elongated shape, to varies wavelength in regard their synchronized wobbling activity. Their results show that the strongest response was observed at blue (430 nm) and far red(780nm) when the diatom phytochrome encoding gene isn't knocked out. They further show that the cell response to blue and far-red light can be inhibited by red light (660nm), while the impact depends on the red to blue light ratio.

The present study is the logical consequence of the authors previous work, where they have established the laser diffractometry allowing them to determine the cell orientation within the water column.

major comments:

L130: Have you tested the ratio impact of FR and R as well? If yes, what were the results? If not tested, why not?

L 170: Why not repeat the experiment on a low cell density culture or on a single cell level?

L: 154 ff: this is a weak point of the hypothesis and should be tested. What is the minimal intensity which a cell can detect and how much do they approximately emit and how close they would need to be to each other in order to be able to receive the FR from one cell to another. How likely is it that it serves as an intracellular signal?

L 189ff. Maybe I simply didn't understand the last experiment, but I have trouble to follow the conclusions. While not pulsed red light seems to inhibit cell wobbling, pulsed red lights seems to trigger it?

Minor comments:

• Fig 1.

o legend title: 'continuous light experiment' is not accurate as it can refer to either wavelength of exposure time. Please rephrase it, e.g. continuous exposure to various wavelengths.

o 1.A and S1 and all following figures. Nice idea to represent the wavelength in the corresponding colors. However, choosing black for FR isn't very intuitive. May you want to change it to a dark red, eg. As in panel 1 C?

o 1A. what rare the dotted and what the solid line stand for? WT and KO? Or is the solid line a fit of the scatterplot? In the latter case:

the fit for 430 and 530 doesn't seem to respect the real shape of the data points. While 430 shows another peak (even though lower) around 350 s 530nm peaks at 150s and not at 200 as the fitted line suggests. While it doesn't contradict the finding that 430 and 780nm result in the highest wobbling response, 430nm also shows a clear interaction. The sentence in L 199 "minimal response elsewhere" thus is not correct and needs to be rephrased accordingly.

o 1B Legend text and figure could be better connected to each other. While the graphs show delta Abs the legend text talks about DPH de-/activation and uses a color-code, which are not defined Green= activated? Blue deactivated? Furthermore, the abbreviations 'Pfr and Pr' are nowhere defined in the MS. Are those data of WT or KO?

o 1C what does Pk/FWHM stand for? Unit? (please define)

• Fig2.

o Legend title: 'continuous' blue and red light. The word doesn't make sense like that. Do you mean continuously changing ratio?

o Panel A&B, highest y axis value cut off

• Fig 4.

- o Panel D. please define the abbreviation within the graph in the figure legend. E.g. What does 5D1 for?
- References:
- o Number 26. Correct title from pennates to pennate
- L446: symbol for micro is not shown
- o The Materials and methods partly “copy paste from: 1st author previous article: Pelagic diatoms communicate through synchronized beacon natural fluorescence signaling (DOI: 10.1126/sciadv.abj5230
- o The used methodology is new to me and seems to be quite sophisticated.
- o The used unit Einstein to express the light intensity is not accepted as an SI unit and outdated, please substitute with μmol quanta (or photons) $\text{m}^{-2} \text{s}^{-1}$
- o The article would benefit from a conceptual diagram to help understand the set-up.
- o I miss some further details:
For how long the cells were the cells exposed to the different wavelength during the different experiments?
L110: How have the authors dealt with the other morphotypes of *Pheodactylum* as the triradiate form?
L113: please define the “two size bands”
L454: ‘This intensity was then modulated sinusoidally with periods of cell wobbling natural frequency ($\sim 140 \text{ s}$)’. It is not explained how the authors did that.

Reviewer #2

(Remarks to the Author)

(review split between senior co-reviewer comments and Early-career co-reviewer comments)

Reviewer #2a (early career co-reviewer):

General comments:

This is a highly interesting and well-structured study. The article is clear and concise, making complex concepts accessible while maintaining scientific rigor. One particularly intriguing question raised is whether DPH mediates cellular responses only to exogenous ambient light or if it is also involved in the perception of autofluorescence. This is a fascinating hypothesis, and the approach used to test it, as well as the results obtained, is particularly interesting. The experimental setup is straightforward and leads to clear results regarding the involvement of phytochromes in the wobbling dance in response to specific wavelengths. However, as explicitly mentioned in the article, further investigation and additional evidence are required to confirm and fully understand the role of intracellular communication. The Materials and Methods section is well described, ensuring good reproducibility. However, more transparency regarding data processing would be beneficial (see individual comments for details).

Three key findings stand out and are clearly highlighted:

- The implication of phytochromes in the synchronization of cell movements.
- The response to red, far-red, and blue light, indicating a qualitative aspect of behavioral signaling.
- The response of phytochromes to autofluorescence signals from diatoms, potentially suggesting intercellular communication.

Overall, this is an innovative and valuable study that contributes to a better understanding of how diatoms use different wavelengths and how they might communicate through light signaling. It opens exciting avenues for future research on photoreception and cellular interactions in microalgae.

Specific comments:

Need for standardization of irradiance units (PPFD):

In the supplementary materials and methods, there is a typo in the "cell culture" paragraph: "and $80 \mu\text{m}^{-2} \text{s}^{-1}$ ". Additionally, for irradiance values, it would be preferable to use the unit " $\mu\text{mol photons}\cdot\text{m}^{-2}\cdot\text{s}^{-1}$ ", which is more commonly used in photophysiology, rather than " $\mu\text{E}\cdot\text{m}^{-2}\cdot\text{s}^{-1}$ ".

The use of the unit "1mW" (appearing multiple times, e.g., in Fig. 3, Fig. 2B, and the supplementary "Fluorescence signal variations measurements") is uncommon, as it refers to power rather than irradiance?

In Figure S1B, why are the ratios not displayed for all wavelengths, as they are in Figure S1A?

I am not entirely sure how Figures 1 and S1 correspond. How are the data transformed to go from Ratios to Power?

The period of cell wobbling natural frequency is given as $\sim 140 \text{ s}$. How was this value determined? Was it measured experimentally on your cells, or is it taken from the literature?

In the main text, Figure S3 is not well integrated and could benefit from additional context. For example:

"Starting with the intensity of B light used in Figure 1, we simulated sinking in the water column by increasing R light (Fig. S3) and observed inhibition of the synchronized wobbling amplitude (Fig. 2), while wobbling frequency remained unaffected (Fig. 2A). »

Provide more transparency on data usage and processing, particularly regarding the number of individuals observed/included for Figures 2A and 4C.

Typo in "autofluoresce" in the Results and Discussion section.

Reviewer #2b (senior reviewer):

The study by Front-Munoz et al. investigates the role of phytochromes in marine diatoms. It is particularly well-conducted, with high-quality and inspiring results. The study clearly demonstrates the role of diatom phytochromes (DPH) in detecting light signals and the behavioral adaptations resulting from this detection in diatoms. The DPH gene appears to be central to the synchronized response of diatoms to light signals. This is, therefore, an excellent study on the perception of light quantity and quality in *Phaeodactylum tricorutum* and the triggering effect of DPH at the community level. That being said, I have some comments on certain aspects of the study that seem unclear to me, as well as others where I believe the results may have been over-interpreted. These points are detailed in the specific comments below.

In general, I find that:

The results are not always well explained in the methodology, making it difficult to read and interpret certain graphs. In addition, the study claims to demonstrate that phytochromes are involved in an intercellular social behavioral response, which would indeed be a novel finding. However, on this second point, I remain skeptical after reading the manuscript. Even though the results are exciting and may seem convincing—particularly Figure 4, which shows diatom responses to pulsed light, I believe there is a lack of direct evidence that diatom autofluorescence indeed serves as a communication signal between cells, driving the observed behaviors. The authors themselves appear to be cautious in their manuscript when establishing this direct link (e.g. L38: potentially mediated, L104: the possibility of endogenous light signal, L150: a plausible explanation, L183: potentially implying, ...). Thus, while the content of the article is not in question, the title could perhaps better reflect its findings.

Specific comments:

L57: It would be nice to define in the introduction the wavelength of the R and FR region.

L110-119 (+figure1): The principle of laser diffractometry is well explained (L110-116), but its connection to Figure 1 is unclear to me. I may have missed something, but unfortunately, the Methods & Materials section in the Supplementary Material does not provide additional clarification. I suggest improving the explanation in the Methods & Materials on how the Power Spectrum (Figure 1A) and the Time Series Ratio (Figure S1A) are related. Additionally, please clarify what Pk/FWHM represents in Figure 1C. It would also be helpful to specify in the Figure 1 caption the differences between the solid and dashed lines—are they meant to represent WT vs. KO?

Lines 136-139: The inclusion of the vertical distribution of the Red:Blue light ratio in the sea (figS3) is a strong point, making the extrapolation of the results to natural environments very convincing.

Lines 147-154: In this section, the authors analyze the synchronized response of diatoms to light signals. It would be helpful to clarify at this stage whether diatom autofluorescence is produced in the R or FR region (or both) when excited in the B, as this information is only provided later (L178, Figure 3). Including it earlier could better support the authors' hypothesis. That being said, in this section, the authors dismiss the possibility of coordinated behavior driven by info-chemicals to introduce the autofluorescence hypothesis. While the autofluorescence hypothesis is intriguing, at this stage of the manuscript, it is sufficient to establish that light acts as a trigger—via DPH—to induce a behavioral response at the community level. Since the hypothesis of autofluorescence as an intercellular signal is introduced later with the pulsed light experiments (L189, Figure 4), it may be premature to bring it up as early as L150.

L154-156: intensity of fluorescence in the sea => This is very interesting and brings the question of the distance at which the autofluorescence of the Chlorophyll a could be detected by other cells in nature. Is *Phaeodactylum tricorutum* widely distributed in several different ecosystems, or rather restricted to clear, oligotrophic or non turbid waters ?

L158-162 & Fig3 : The manuscript states that the measured chlorophyll fluorescence includes both R and FR components. However, in Figure 3, it is mentioned that the Red fluorescence corresponds to a long-pass filter that collects light >645 nm. This implies that the R signal actually represents R + FR, while the FR signal refers only to FR wavelengths (> 715nm). If this is correct, it makes sense that the emitted R+FR and FR signals produce similar results, as the R+FR signal would be largely dominated by FR fluorescence.

L180-181: The wobbling motility of diatoms is described as "incoherent" in dph-KO cells, but this point needs further clarification. When comparing the power spectrum curves in Figure 4A&B (pulsed light) with Figure 1A (continuous light), I don't see a significant difference. What are the main distinctions? What exactly is considered "incoherent" here? The authors then emphasize that DPH is essential for triggering collective behavior(which the study clearly demonstrates). However, they also reintroduce the hypothesis of autofluorescence as an intercellular signal. This needs a more detailed explanation because it is not immediately clear how Figure 4A&B supports the autofluorescence hypothesis.

L189-197: From my perspective, this paragraph is the only one that supports the autofluorescence hypothesis. It is indeed intriguing that R light does not induce movement when provided continuously (Fig1C) but does when introduced at a period matching the wobbling frequency (Fig4C). This serves as an argument in favor of the autofluorescence hypothesis, even

though the direct relationship between autofluorescence and movement is not explicitly demonstrated in this case. Thus, I suggest presenting this result earlier in the manuscript.

L221: "strong evidence of the relevance of light signaling (remove communication)"

M&M: Continuous light experiments were conducted by exposing the cells to different wavelengths for 2 hours. Meanwhile, laser diffractometry measurements—central to establishing time series ratios and power spectra—were performed using a LISST-100X, which operates at 670 nm (R light). Looking at Figures S1 and 2, it appears that the measurement duration exceeds 3000 seconds (i.e., more than 50 minutes). This raises the question of how the LISST-100X, which exposes the cells to red light for a relatively long period, aligns with the experimental treatments that are supposed to provide only blue and FR light. Could you clarify how this potential interference is accounted for?

Reviewer #3

(Remarks to the Author)

Version 2:

Reviewer comments:

Reviewer #2

(Remarks to the Author)

(review split between senior co-reviewer comments and Early-career co-reviewer comments)

Reviewer #2a (early career co-reviewer):

My comments have been considered, and the revised version of the manuscript generally addresses the critiques raised during the first round, which improves the clarity, methodological rigor, and overall structure of the manuscript. However, some imprecisions remain.

General comments:

There are numerous typographical errors throughout the document, particularly missing spaces between numerical values and their units. While spaces are sometimes used correctly, they are frequently omitted, and the manuscript requires consistent homogenization.

There is a lack of transparency regarding the number of biological replicates used to compute means and standard deviations throughout the manuscript. This information is not provided in the supplementary Materials and Methods section nor in the figure legends. The authors should clearly the number of replicates ($n = ?$) used for the following figures: Figure 1C, 2A, 4C, 4D, S2A, S2B, S5A, and S5B.

Specific comments:

L113: The detailed technical description of the two size bands (7.33–14 μm and 2.5–3.5 μm) used in the laser diffractometry (LD) measurement seems superfluous for the Results and Discussion section, and the mention of "two size bands" is somewhat redundant with the following sentence L116. Wouldn't it be more appropriate to move this explanation to the supplementary Materials and Methods section on laser diffractometry? Doing so would help create a more concise and focused paragraph?

Figure 1A: I still have some uncertainties regarding the application of a Gaussian fit to the full spectrum when no single peak is clearly dominant ($\text{std}(\text{Pro}) < 1$). Is this approach truly justified? What is its relevance and robustness?

Moreover, is there a statistical or physiological justification for using the $\text{std}(\text{Pro}) > 1$ threshold as a criterion for this data?

Finally, it is good that arrows for both FWHM and Pk were added to the graph to improve readability. However, the arrow for Pk is not visually optimal, as it appears somewhat floating within the graph. Although it is mentioned in the legend, it would be better to find a clearer visual solution to link the "Pk" label to its arrow, similar to the "FWHM" arrow.

Typo:

L173: « autofluoresce » instead autofluorescence

L224 and 229: « signalling » instead signaling

L429: « cells/L » unit instead cells mL^{-1}

L511: « impa » instead impact

Figure 4A: add space between period and (s) on x axis

Reviewer #2b (senior co-reviewer):

After thoroughly reviewing the revised manuscript, I can confirm that all of my previous comments and suggestions have been satisfactorily addressed by the authors. I also fully agree with the observations and recommendations made by my co-reviewer.

Regarding Reviewer #1's comments on the experimental design—specifically the impact of the FR and R ratios, which were previously identified as a weak point in the study—I find that the authors have now addressed this concern convincingly. They provided new data demonstrating the effects of varying FR-to-R ratios, and the results show a behavior similar to that observed with the B/R ratio. This addition strengthens the overall validity of the experimental design.

At this stage, I have no further comments or suggestions concerning the manuscript.

However, I remain curious about 2 points :

1- In the continuous light experiment (Fig. 1A), the power spectrum of the ratio still shows a weaker, yet seemingly significant, signal in the green, orange, and red wavelength ranges, with a longer period (in seconds). To me, this suggests that the “wobbling” becomes weaker and occurs at a higher frequency as one moves from the blue toward the red region of the light spectrum (with the exception of FR of course). I am wondering whether the signals recorded between approximately 200 and 400 seconds (depending on the wavelength) correspond to different wobbling frequencies, or whether they might instead result from the Brownian motion of the cells. In the latter case, would one expect such an effect to vary with wavelength?

2- In line with this question, regarding Figure 4 (“Pulsed light experiments”): does the fact that the authors selected a pulsing frequency of approximately 150 s—which corresponds to the main signal observed under continuous B and FR light—introduce a potential element of circular reasoning? I am inclined to think not, since Figure 4A indeed shows a power spectrum peak around 150 s under pulsed R light. However, I would appreciate confirmation of this interpretation, or at least clarification on how to interpret the weaker signals observed at higher frequencies.

Overall, this study is excellent and highly stimulating, although at times somewhat challenging to follow—even though the authors have made substantial improvements compared with the previous version. The methodology is very specific and complex, and the constraints of the manuscript's length make it difficult to provide a fully comprehensive description without referring the reader to earlier publications where the methods are explained in greater detail. While this approach is understandable, it does make the reader's (and reviewer's) task more demanding, as one must frequently cross-reference multiple papers to fully grasp the experimental procedures—sometimes leaving the impression that one must take certain methodological aspects on trust rather than direct verification.

That said, at this stage, I do not consider it necessary for the authors to include additional methodological details in the current version of the manuscript. Perhaps, I could invite them to pay extra attention with the notation throughout the manuscript. For instance, Peak amplitude of the power spectrum is noted "Pk" in figure 1A and $P(\omega)$ in fig4C,D.

Reviewer #3

(Remarks to the Author)

Version 3:

Reviewer comments:

Reviewer #2

(Remarks to the Author)

After thoroughly reviewing the revised manuscript, I (and the early-career researcher) can confirm that all of our previous comments and suggestions have been satisfactorily addressed by the authors.

RESPONSE TO REVIEWERS' COMMENTS

The authors tested the cell response of *Pheodactylum tricornutum* as a representative for pelagic diatoms with an elongated shape, to varies wavelength in regard their synchronized wobbling activity. Their results show that the strongest response was observed at blue (430 nm) and far red(780nm) when the diatom phytochrome encoding gene isn't knocked out. They further show that the cell response to blue and far-red light can be inhibited by red light (660nm), while the impact depends on the red to blue light ratio.

The present study is the logical consequence of the authors previous work, where they have established the laser diffractometry allowing them to determine the cell orientation within the water column.

major comments:

L130: Have you tested the ratio impact of FR and R as well? If yes, what were the results? If not tested, why not?

In the first version of the manuscript, we analyzed the impact of B/R ratios as the most relevant ones in the marine environment (associated for instance with varying depth). The Reviewer 1 is right that, if light-induced synchronized wobbling is mediated by DPH, the induction by FR should also be counteracted by red light. As suggested by Reviewer 1, we have included new data also showing the impact of varying FR to R ratios. The results indicate a similar behavior to that shown in Fig2b; i.e. a decaying oscillation amplitude with lower FR to R ratio. The analysis confirms the involvement of the diatom phytochrome photo perception in the analyzed processes, and is coherent with the mode of action of DPH recently described in Duchene et al. (2025)). The new data are now included in figure S4 in the Sup. Mat. In the text, we have added the following sentence to line 133: "a similar response was observed in experiments performed with varying FR to R ratios".

L: 154 ff: this is a weak point of the hypothesis and should be tested.

'The capacity of diatoms to detect and respond to weak but characteristic fluorescence emission of neighboring cells is already demonstrated in the work of Liu et al. (2021), where this signal is shown to trigger cellular responses such as algal iron uptake'. We expand on this in the following comments.

What is the minimal intensity which a cell can detect?

We did not test the minimal intensity required to trigger wobbling. Photoreceptors are in fact capable of detecting single-photons (see Hegemann 2008), a common feature of many photochemical processes. However, it is indeed an interesting question whether there is a minimum intensity threshold (or cell-to-cell distance) to initiate, after detection, a signal transduction pathway leading to a measurable cellular photoresponse. Duchene et al measure that phytochrome-dependent gene expression changes are induced by intensities of $1\mu\text{mol photon/m}^2/\text{s}$, or even lower ($0.1\mu\text{mol/m}^2/\text{s}$) under blue light. From Liu et al. (2021) (see their figures 5 and 6) we also know that's the case for endogenously emitted chlorophyll autofluorescence with emission intensities probed on populations densities similar to those explored in our work. Although we cannot yet pinpoint the actual value of that threshold for the triggering of cell oscillations, we can at the very least provide an upper bound based on those previous observations.

how much do they approximately emit? Measured fluorescence emission at the used cell concentration is shown in Fig. 3A and correspond to an intensity about $10^{(-4)}$ that used for excitation. This is also consistent with previous measurements By Liu et al. (2021).

how close they would need to be to each other in order to be able to receive the FR from one cell to another?

As previously mentioned, light perception occurs at the single-photon level and does not depend on the distance to the source. What may depend on the distance between cells is whether the perceived signal is sufficient to elicit a photoresponse. We estimate a maximum effective separation of approximately 100 μm , typical of dense suspensions. Given a typical fluorescence yield of 1%, the fluorescence emitted by a single neighboring cell would contribute approximately 0.0002–0.00007 $\mu\text{mol m}^{-2} \text{s}^{-1}$. However, this is a collective process, and a cell receives fluorescent light from a surrounding constellation of neighboring cells, as illustrated in the schematic below (adapted from Liu et al.). When this collective effect is taken into account, the light intensity at a distance of 100 μm can reach values on the order of 0.1 $\mu\text{mol m}^{-2} \text{s}^{-1}$.

Fig. Schematic illustration of the mathematical model for estimating the chlorophyll fluorescence photon flux density (CFPFD) received per cell. (a) Two adjacent cells. (b) Two cells with cellular distance at 10 μm . (c) Two cells with cellular distance at 100 μm . Schematic illustration of the mathematical model for estimating the chlorophyll fluorescence photon flux (CFPF) during diatom blooms in oceans. (a) Evenly distributed cells in real water. The cell in the center can receive chlorophyll fluorescence from the whole space. (b) Percentage of CFPF versus upper limit of the integral.

How likely is it that it serves as an (inter)cellular signal?

In the manuscript, we do not taxatively state that fluorescence serves as “the” intercellular signal, but rather propose it is a plausible signal. We consider that the cumulative evidence provided by the experimental results shown here, together with previously published results, supports this statement:

- i) Photoresponse spectra partially overlap with that of natural autofluorescence.
- ii) For RL, there is a strong photoresponse for time-periodic excitation with periods close to those of the natural cell wobbling.
- iii) Experiments like those performed by Liu et al. (2021) demonstrate the cells' capability to respond to fluorescence emissions by neighbouring cells.

Unfortunately, direct evidence is not currently available as it would require experiments where cell fluorescence emission is blocked while perception is maintained, which remains technically challenging in a cell population.

L 170: Why not repeat the experiment on a low cell density culture or on a single cell level?

The sentence could be interpreted to refer to a density-dependent signal, which is not the case. Performing experiments at different cell densities wouldn't solve the technical challenge previously referred to, as emission and perception aren't easily decoupled. Moreover, all the potential coupling factors (hydrodynamic coupling, chemical signaling, and auto-fluorescence signaling) would depend on the average distance between cells (i.e., density); such an experiment would therefore not help discriminate between the different coupling factors. We have modified the sentence to avoid misinterpretations. It now reads: "Exogenous light and chlorophyll-emitted light signals cannot easily be decoupled."

L 189ff. Maybe I simply didn't understand the last experiment, but I have trouble to follow the conclusions. While not pulsed red light seems to inhibit, pulsed red lights seems to trigger it? This is correctly understood. The experiments indeed show that pulsed red light (RL) at frequencies comparable to natural wobbling (periods ~100 s) elicits a strong photoresponse. In contrast, cells do not exhibit a photoresponse under continuous RL exposure. This observation supports the hypothesis that cell autofluorescence may act as an intercellular signal. It is also consistent with previous findings on cell wobbling under non-stationary illumination (see Font-Muñoz et al., Sci. Adv., 2021). The mechanisms underlying the responses to continuous and pulsed light may involve distinct pathways, as we now emphasize in the main text.

Minor comments:

- Fig 1. In to legend title: 'continuous light experiment' is not accurate as it can refer to either wavelength of exposure time. Please rephrase it, e.g. continuous exposure to various wavelengths. **Done.**

- o 1.A and S1 and all following figures. Nice idea to represent the wavelength in the corresponding colors. However, choosing black for FR isn't very intuitive. May you want to change it to a dark red, eg. As in panel 1 C? **Done.**

- o 1A. what are the dotted and what the solid line stand for? WT and KO? Or is the solid line a fit of the scatterplot? **The latter.**

In the latter case: the fit for 430 and 530 doesn't seem to respect the real shape of the data points. While 430 shows another peak (even though lower) around 350 s 530nm peaks at 150s and not at 200 as the fitted line suggests. While it doesn't contradict the finding that 430 and 780nm result in the highest wobbling response, 530nm also shows a clear interaction. The sentence in L 119 "minimal response elsewhere" thus is not correct and needs to be rephrased accordingly.

While we agree with the reviewer that there are indeed multiple peaks in all cases, we have used a standard and robust criteria to discriminate and characterize the main component of each power spectrum. The criteria is based on conventional signal processing tools designed to locate peaks and arrange them by their relative prominence (Pro) in the signal. For signals with a variance of the relative peak prominence above one, $\text{std}(\text{Pro}) > 1$ (i.e. when a main peak can clearly be isolated), we stick to a Gaussian fit to the main peak. For $\text{std}(\text{Pro}) < 1$ (i.e. when no peak stands out in a statistically significant way above the others) a single Gaussian is fitted across the full spectrum. Although a refined analysis of the secondary peaks might be of eventual interest, we do not consider that it has an impact on the results presented in this manuscript, so we defer this to future work. This is now better explained in the Materials and Methods section of the Supplementary Material where we also include the values of "peak prominence" for each case.

o 1B Legend text and figure could be better connected to each other. While the graphs show delta Abs the legend text talks about DPH de-/activation and uses a color-code, which are not defined Green= activated? Blue deactivated? Furthermore, the abbreviations 'Pfr and Pr' are nowhere defined in the MS. **We have now modified the figure caption to correct for all these discrepancies.**

Are those data of WT or KO? **WT**

o 1C what does Pk/FWHM stand for? Unit? (please define) **Pk/FWHM stand for peak amplitude (Pk) of the power spectrum (shown in fig 1A) normalized by its full width at half maximum (FWHM). These are given in arbitrary units. To help the readers we now showcase an example of both variables for one of the power spectra in panel A.**

• Fig2.

o Legend title: 'continuous' blue and red light. The word doesn't make sense like that. Do you mean continuously changing ratio? **The referee is right as this, once again, refers to "continuous exposure". We have modified the text to reflect that.**

o Panel A&B, highest y axis value cut off. **That is true and it has been corrected in the resubmitted version of the figure.**

• Fig 4.

o Panel D. please define the abbreviation within the graph in the figure legend. E.g. What does 5D1 for? **These abbreviations correspond to the different strains used in our experiments. We have now clarified this in the caption.**

• References:

o Number 26. Correct title from pennates to pennate. **Done.**

• L446: symbol for micro is not shown. **Done.**

o The Materials and methods partly "copy paste from: 1st author previous article: Pelagic diatoms communicate through synchronized beacon natural fluorescence signaling (DOI: 10.1126/sciadv.abj5230) **We have now shorten the section and refer the readers to Font-Muñoz et al. Sci. Adv. (2021).**

o The used methodology is new to me and seems to be quite sophisticated.

o The used unit Einstein to express the light intensity is not accepted as an SI unit and outdated, please substitute with $\mu\text{mol quanta (or photons) m}^{-2} \text{ s}^{-1}$. **Done.**

o The article would benefit from a conceptual diagram to help understand the set-up.

ADD A modified version in supp here again to make the reviewer HAPPY and simplify the reading!!

The experimental set-up used in this work was developed by our group a few years ago and was already used in previous work. We refer the interested readers to Fig. S1 in Font-Muñoz et al. Sci. Adv. (2021). Differences rely on the use of different excitation spectral bands and phytoplankton strains. This is now commented in the Materials and Methods section of the Supplementary Materials.

o I miss some further details:

For how long were the cells exposed to the different wavelength during the different experiments? **Cells were exposed to light for 2 hours. This information is now available in the Materials and Methods section of the Supplementary Materials.**

L110: How have the authors dealt with the other morphotypes of *Pheodactylum* as the triradiate form?
Our laboratory cultures are maintained under conditions that exclusively promote the fusiform morphotype. Fusiform cells are typically obtained by culturing under white light and maintaining the cultures in the exponential growth phase.

L113: please define the “two size bands” Done.

L454: ‘This intensity was then modulated sinusoidally with periods of cell wobbling natural frequency (~140 s)’. It is not explained how the authors did that. We coupled the controller triggering the excitation LED source to a pulse generator. This is now described in the Materials and Methods section of the Supplementary Materials.

Reviewer #2 (Remarks to the Author):

(review split between senior co-reviewer comments and Early-career co-reviewer comments)

- Reviewer #2a (early career co-reviewer):

General comments:

This is a highly interesting and well-structured study. The article is clear and concise, making complex concepts accessible while maintaining scientific rigor. One particularly intriguing question raised is whether DPH mediates cellular responses only to exogenous ambient light or if it is also involved in the perception of autofluorescence. This is a fascinating hypothesis, and the approach used to test it, as well as the results obtained, is particularly interesting. The experimental setup is straightforward and leads to clear results regarding the involvement of phytochromes in the wobbling dance in response to specific wavelengths. However, as explicitly mentioned in the article, further investigation and additional evidence are required to confirm and fully understand the role of intracellular communication. The Materials and Methods section is well described, ensuring good reproducibility. However, more transparency regarding data processing would be beneficial (see individual comments for details).

Three key findings stand out and are clearly highlighted:

- The implication of phytochromes in the synchronization of cell movements.
- The response to red, far-red, and blue light, indicating a qualitative aspect of behavioral signaling.
- The response of phytochromes to autofluorescence signals from diatoms, potentially suggesting intercellular communication.

Overall, this is an innovative and valuable study that contributes to a better understanding of how diatoms use different wavelengths and how they might communicate through light signaling. It opens exciting avenues for future research on photoreception and cellular interactions in microalgae.

We thank the reviewer for the general remarks on the manuscript and address the specific comments below.

Specific comments:

Need for standardization of irradiance units (PPFD):

In the supplementary materials and methods, there is a typo in the "cell culture" paragraph: "and 80 $\mu\text{m}^{-2}\text{s}^{-1}$ ". Additionally, for irradiance values, it would be preferable to use the unit " $\mu\text{mol photons}\cdot\text{m}^{-2}\cdot\text{s}^{-1}$ ", which is more commonly used in photophysiology, rather than " $\mu\text{E}\cdot\text{m}^{-2}\cdot\text{s}^{-1}$ ". » **Done.**

The use of the unit "1mW" (appearing multiple times, e.g., in Fig. 3, Fig. 2B, and the supplementary "Fluorescence signal variations measurements") is uncommon, as it refers to power rather than irradiance? **We agree with the referee and have now transformed all units to irradiance.**

In Figure S1B, why are the ratios not displayed for all wavelengths, as they are in Figure S1A? **Experiments with dph knockouts were only performed for the three wavelengths for which a photoresponse in the WT was observed.**

I am not entirely sure how Figures 1 and S1 correspond. How are the data transformed to go from Ratios to Power? **The power spectra shown in fig. 1A are estimated using fast Fourier transform (FFT) of the time series of the Ratios shown in fig. S1**

The period of cell wobbling natural frequency is given as ~ 140 s. How was this value determined? Was it measured experimentally on your cells, or is it taken from the literature? **The wobbling natural frequency was estimated from our experiments by locating the position of the main peak in the power spectra presented in Fig. 3B (also now indicated for the power spectra shown in fig 1A)**

In the main text, Figure S3 is not well integrated and could benefit from additional context. For example:

"Starting with the intensity of B light used in Figure 1, we simulated sinking in the water column by increasing R light (Fig. S3) and observed inhibition of the synchronized wobbling amplitude (Fig. 2), while wobbling frequency remained unaffected (Fig. 2A). » **Done.**

Provide more transparency on data usage and processing, particularly regarding the number of individuals observed/included for Figures 2A and 4C. **The bars in figure 2A show the mean plus/minus standard deviation of all individual experiments (n=4) performed with varying irradiance (shown in panel B). Those in figure 4D also show the mean plus/minus standard deviation of all performed experiments with either responsive (WT and Tc) or knockout strains, respectively.**

Typo in "autofluoresce" in the Results and Discussion section. **Corrected.**

- Reviewer #2b (senior reviewer):

The study by Front-Munoz et al. investigates the role of phytochromes in marine diatoms. It is particularly well-conducted, with high-quality and inspiring results. The study clearly demonstrates the role of diatom phytochromes (DPH) in detecting light signals and the behavioral adaptations resulting from this detection in diatoms. The DPH gene appears to be central to the synchronized response of diatoms to light signals. This is, therefore, an excellent study on the perception of light quantity and quality in *Phaeodactylum tricorutum* and the triggering effect of DPH at the community level. That being said, I have some comments on certain aspects of the study that seem unclear to me, as well as others where I believe the results may have been over-interpreted. These points are detailed in the specific comments below.

We thank the reviewer for the general remarks on the manuscript and address the specific comments below.

In general, I find that:

The results are not always well explained in the methodology, making it difficult to read and interpret certain graphs. In addition, the study claims to demonstrate that phytochromes are involved in an intercellular social behavioral response, which would indeed be a novel finding. However, on this second point, I remain skeptical after reading the manuscript. Even though the results are exciting and may seem convincing—particularly Figure 4, which shows diatom responses to pulsed light, I believe there is a lack of direct evidence that diatom autofluorescence indeed serves as a communication signal between cells, driving the observed behaviors.

In the manuscript, we do not taxatively state that fluorescence serves as “the” intercellular signal, but rather propose it is a plausible signal. We consider that the cumulative evidence provided by the experimental results shown here, together with previously published results, supports this statement:

- i) Photoresponse spectra partially overlap with that of natural autofluorescence.
- ii) For RL, there is a strong photoresponse for time-periodic excitation with periods close to those of the natural cell wobbling.
- iii) Experiments like those performed by Liu et al. (2021) demonstrate the cells' capability to respond to fluorescence emissions by neighbouring cells.

Unfortunately, direct evidence is not currently available as it would require experiments where cell fluorescence emission is blocked while perception is maintained, which remains technically challenging in a cell population.

The authors themselves appear to be cautious in their manuscript when establishing this direct link (e.g. L38: potentially mediated, L104: the possibility of endogenous light signal, L150: a plausible explanation, L183: potentially implying, ...). Thus, while the content of the article is not in question, the title could perhaps better reflect its findings.

We respectfully disagree with the referee on this point. The current title does not suggest the direct link between cell autofluorescence and collective behaviour. Instead, it focuses on the role of DPH in collective responses, which we believe it is well established by our findings.

Specific comments:

L57: It would be nice to define in the introduction the wavelength of the R and FR region. **Done. We have now included it in the second paragraph of the introduction.**

L110-119 (+figure1): The principle of laser diffractometry is well explained (L110-116), but its connection to Figure 1 is unclear to me. I may have missed something, but unfortunately, the Methods & Materials section in the Supplementary Material does not provide additional clarification. I suggest improving the explanation in the Methods & Materials on how the Power Spectrum (Figure 1A) and the Time Series Ratio (Figure S1A) are related. **We have modified the Methods section now to include further details on our estimates of the power spectra of the Ratio signals.**

Additionally, please clarify what Pk/FWHM represents in Figure 1C. **Done. We have modified fig 1A and the figure caption to indicate what Pk (peak height) and FWHM (full width at half maximum) stand for.**

It would also be helpful to specify in the Figure 1 caption the differences between the solid and dashed lines—are they meant to represent WT vs. KO? **There are no dashed lines in fig1. We guess the referee is**

thinking about the dotted lines in panel A, which correspond to the actual data points; the solid lines instead are gaussian fits to those points, as indicated now in the modified caption.

Lines 136-139: The inclusion of the vertical distribution of the Red:Blue light ratio in the sea (figS3) is a strong point, making the extrapolation of the results to natural environments very convincing. We thank the reviewer for this appreciation of the relevance of our findings to natural environments.

Lines 147-154: In this section, the authors analyze the synchronized response of diatoms to light signals. It would be helpful to clarify at this stage whether diatom autofluorescence is produced in the R or FR region (or both) when excited in the B, as this information is only provided later (L178, Figure 3). Including it earlier could better support the authors' hypothesis. As suggested by the reviewer, we now mention in this paragraph the wavelength bands of the emitted autofluorescence.

That being said, in this section, the authors dismiss the possibility of coordinated behavior driven by infochemicals to introduce the autofluorescence hypothesis. While the autofluorescence hypothesis is intriguing, at this stage of the manuscript, it is sufficient to establish that light acts as a trigger—via DPH—to induce a behavioral response at the community level. Since the hypothesis of autofluorescence as an intercellular signal is introduced later with the pulsed light experiments (L189, Figure 4), it may be premature to bring it up as early as L150.

The focus of our manuscript is on the relevance of DPH in triggering collective behavior. Our experiments firmly demonstrate this role. We further speculate that fluorescence light emission and unknown component downstream of DPH signalling could be involved in the measured photoresponse. That said, we want to stress we do not intend to discard either physical disturbances or dissolved infochemicals as potential cell-to-cell coupling signals, but rather hypothesize fluorescence could be an additional interaction mechanism compatible with all current observations. As already mentioned above, we cannot directly test this hypothesis, but we can provide additional evidence (including some new experimental results; e.g. figure 4) that supports it. Based on this argument, we believe the reader would better discriminate between speculative discussion and proven facts with the current logical structure.

L154-156: intensity of fluorescence in the sea => This is very interesting and brings the question of the distance at which the autofluorescence of the Chlorophyll a could be detected by other cells in nature.

Light perception occurs at the single-photon level and does not depend on the distance to the source. What may depend on the distance between cells is whether the perceived signal is sufficient to elicit a photoresponse. We estimate a maximum effective separation of approximately 100 μm, typical of dense suspensions. Given a typical fluorescence yield of 1%, the fluorescence emitted by a single neighboring cell would contribute approximately 0.0002–0.00007 μmol m⁻² s⁻¹. However, this is a collective process, and a cell receives fluorescent light from a surrounding constellation of neighboring cells, as illustrated in the schematic below (adapted from Liu et al.). When this collective effect is taken into account, the light intensity at a distance of 100 μm can reach values on the order of 0.1 μmol m⁻² s⁻¹.

Fig. Schematic illustration of the mathematical model for estimating the chlorophyll fluorescence photon flux density (CFPFD) received per cell. (a) Two adjacent cells. (b) Two cells with cellular distance at 10 μm . (c) Two cells with cellular distance at 100 μm . Schematic illustration of the mathematical model for estimating the chlorophyll fluorescence photon flux (CFPF) during diatom blooms in oceans. (a) Evenly distributed cells in real water. The cell in the center can receive chlorophyll fluorescence from the whole space. (b) Percentage of CFPF versus upper limit of the integral.

Is *Phaeodactylum tricornutum* widely distributed in several different ecosystems, or rather restricted to clear, oligotrophic or non turbid waters? *P. tricornutum*, is a marine pennate diatom commonly found in coastal waters, including highly productive areas like estuaries, rock pools, and shallow waters exposing the species to important fluctuations in light intensity and salinity.

L158-162 & Fig3 : The manuscript states that the measured chlorophyll fluorescence includes both R and FR components. However, in Figure 3, it is mentioned that the Red fluorescence corresponds to a long-pass filter that collects light $>645\text{ nm}$. This implies that the R signal actually represents R + FR, while the FR signal refers only to FR wavelengths ($>715\text{ nm}$). If this is correct, it makes sense that the emitted R+FR and FR signals produce similar results, as the R+FR signal would be largely dominated by FR fluorescence.

We agree with the reviewer that what we show in figure 3A is either the FR or R+FR components of the emitted signal. However, while the frequency of the measured oscillations is similar, note that the amplitudes are not (see left and right vertical axis range). In particular, the R+FR signal (i.e. $>645\text{ nm}$) doubles in amplitude the FR signal ($>715\text{ nm}$), clearly demonstrating the coherent contribution of the R signal to the overall emission.

L180-181: The wobbling motility of diatoms is described as "incoherent" in dph-KO cells, but this point needs further clarification. When comparing the power spectrum curves in Figure 4A&B (pulsed light) with Figure 1A (continuous light), I don't see a significant difference. What are the main distinctions? While in figure 1A each power spectrum corresponds to a distinct excitation wavelength, all curves shown in fig 4A (or correspondingly in 4B) represent the power spectrum for a single excitation wavelength (FR or R, respectively) but different strains (WT, KOs and Tc). Although the power spectra for WT and Tc strains are indeed similar to that of Figure 1 for FR light, it is not the case for R light: the WT line shows a clear peak at $\sim 140\text{ s}$ in pulse R but not in continuous red light (Fig 1). The power spectrum of the WT in continuous R (Fig 1, dashed red line) rather resembles that of DPH KO mutants under pulse R light (dashed lines in Fig. 4B).

What exactly is considered "incoherent" here? A lack of a clear peak in the power spectrum of the oscillatory signal. That's the case for certain excitation wavelength for the WT strain under continuous exposure (fig 1A) but also for the knock-outs under pulsed light excitation in the R and FR (fig 4A & B).

The authors then emphasize that DPH is essential for triggering collective behavior (which the study clearly demonstrates). However, they also reintroduce the hypothesis of autofluorescence as an intercellular signal. This needs a more detailed explanation because it is not immediately clear how Figure 4A&B supports the autofluorescence hypothesis.

The experiments performed under pulsed excitation light, the results of which are presented in fig 4, are intended to mimic the response of a population to an oscillatory fluorescence signal at the natural characteristic frequency of cell wobbling. As the measured R and FR autofluorescence emissions oscillate with a well-defined frequency (see fig 3), we interrogated cells for their photoresponse to exogenous light

with the same characteristics. The results presented in Figure 4 show that a clear photoresponse exists in this case, supporting the hypothesis that fluorescence emission can trigger collective behavior mediated either directly by DPH or indirectly through an unknown component downstream of DPH signalling. We have now expanded the paragraph to explain more thoroughly the connection between these experiments and this hypothesis.

L189-197: From my perspective, this paragraph is the only one that supports the autofluorescence hypothesis. It is indeed intriguing that R light does not induce movement when provided continuously (Fig1C) but does when introduced at a period matching the wobbling frequency (Fig4C). This serves as an argument in favor of the autofluorescence hypothesis, even though the direct relationship between autofluorescence and movement is not explicitly demonstrated in this case. Thus, I suggest presenting this result earlier in the manuscript.

We agree with the reviewer but decided to remain cautious about the mechanisms involved in the communication process. While we strongly believe there are several arguments pointing to the relevance of cell autofluorescence in this process, direct proof is still lacking. This by no means belittles the direct proof we do provide on the relevance of DPH for collective behaviour. Hence, we would prefer to keep this as the main focus of our manuscript while keeping the light-mediated interaction as a working hypothesis.

L221: "strong evidence of the relevance of light signaling (remove communication)" Done.

M&M: Continuous light experiments were conducted by exposing the cells to different wavelengths for 2 hours. Meanwhile, laser diffractometry measurements—central to establishing time series ratios and power spectra—were performed using a LISST-100X, which operates at 670 nm (R light). Looking at Figures S1 and 2, it appears that the measurement duration exceeds 3000 seconds (i.e., more than 50 minutes). This raises the question of how the LISST-100X, which exposes the cells to red light for a relatively long period, aligns with the experimental treatments that are supposed to provide only blue and FR light. Could you clarify how this potential interference is accounted for?

The laser from the LISST-100X diffractometer operates by pulsing short measuring pulses synchronized with the sensors' electronics. These pulses, although emitting in the R as correctly pointed out by the reviewer, are of extremely short duration compared to natural wobbling periods (ms vs. ~100s). We have measured no photoresponse from cells exposed only to this measuring light. Moreover, even in the unlikely event that there might be a weak response below our sensitivity, this induced photoresponse would be present as a background signal for all our experiments and would not be relevant for the measured relative responses and the extracted conclusions.

RESPONSE TO REVIEWERS' COMMENTS

Reviewer #2 (Remarks to the Author): (review split between senior co-reviewer comments and Early-career co-reviewer comments)

Reviewer #2a (early career co-reviewer):

My comments have been considered, and the revised version of the manuscript generally addresses the critiques raised during the first round, which improves the clarity, methodological rigor, and overall structure of the manuscript. However, some imprecisions remain.

General comments:

There are numerous typographical errors throughout the document, particularly missing spaces between numerical values and their units. While spaces are sometimes used correctly, they are frequently omitted, and the manuscript requires consistent homogenization.

We thank the reviewer for the positive feedback and constructive remarks. We have carefully revised the manuscript to correct all typographical inconsistencies, including missing spaces between numerical values and units, which have now been homogenized throughout.

There is a lack of transparency regarding the number of biological replicates used to compute means and standard deviations throughout the manuscript. This information is not provided in the supplementary Materials and Methods section nor in the figure legends. The authors should clearly the number of replicates ($n = ?$) used for the following figures: Figure 1C, 2A, 4C, 4D, S2A, S2B, S5A, and S5B.

We thank the reviewer for noting this omission. The number of replicates is now indicated in all relevant figure legends and in the Supplementary Materials and Methods section.

Specific comments:

L113: The detailed technical description of the two size bands (7.33–14 μm and 2.5–3.5 μm) used in the laser diffractometry (LD) measurement seems superfluous for the Results and Discussion section, and the mention of “two size bands” is somewhat redundant with the following sentence L116. Wouldn't it be more appropriate to move this explanation to the supplementary Materials and Methods section on laser diffractometry? Doing so would help create a more concise and focused paragraph?

We thank the reviewer for this suggestion. The detailed description of the two size bands has been removed from the Results and Discussion and moved to the

Materials and Methods section to improve clarity and avoid redundancy.

Figure 1A: I still have some uncertainties regarding the application of a Gaussian fit to the full spectrum when no single peak is clearly dominant ($\text{std}(\text{Pro}) < 1$). Is this approach truly justified? What is its relevance and robustness?

We thank the reviewer for this constructive and insightful comment. The criterion based on the standard deviation of peak prominence ($\text{std}(\text{Pro}) > 1$) was selected as a simple, quantitative, and statistically robust approach to determine whether a single oscillation mode dominates the power spectrum.

Peak prominence (Pro) quantifies the relative contrast of each spectral peak with respect to its neighboring minima, enabling reliable identification of dominant, isolated modes. In our dataset, cases where $\text{std}(\text{Pro}) > 1$ consistently correspond to spectra with a clearly dominant peak, whereas $\text{std}(\text{Pro}) < 1$ indicates that multiple peaks contribute comparably to the overall signal. In the latter case, a Gaussian fit over the full spectrum is therefore appropriate to capture its broadband character.

This approach is further supported by an independent, complementary criterion—identifying the dominant peak as an outlier relative to the distribution of all peak prominences ($P_{\text{max}} > \text{mean}(\text{Pro}) + 2 \cdot \text{std}(\text{Pro})$). Both criteria yielded identical classifications across all spectra, robustly distinguishing single-peak from multi-peak or flatter spectral profiles.

Accordingly, applying a Gaussian fit when $\text{std}(\text{Pro}) < 1$ is statistically justified and accurately reflects the underlying signal structure, as further confirmed by agreement with the outlier-based metric. Importantly, as shown in Figures R1 and R2, the overall results remain consistent regardless of whether this criterion is applied. Even when spectra are fitted uniformly with either a single Gaussian or multiple Gaussians, a clear separation persists between responses under blue and far-red illumination (which show strong signals) and those at other wavelengths (where the responses are markedly weaker or absent).

Figure R1. A) Power spectra and their corresponding Gaussian fits obtained by fitting each spectrum with a single Gaussian. B) Ratio between the height of the Gaussian and its FWHM (full width at half maximum).

Figure R2 A) Power spectra and their corresponding Gaussian fits obtained by fitting each spectrum with multiple Gaussians; the highest Gaussian is highlighted in the figure. B) Ratio between the height of the highest Gaussian in each fit and its FWHM (full width at half maximum).

Moreover, is there a statistical or physiological justification for using the $\text{std}(\text{Pro}) > 1$ threshold as a criterion for this data?

We thank the reviewer for this point. The threshold $\text{std}(\text{Pro}) > 1$ is data-driven and was validated against an independent outlier-based criterion ($P_{\text{max}} > \text{mean}(\text{Pro}) + 2 * \text{std}(\text{Pro})$), which produced identical classifications across all spectra.

Finally, it is good that arrows for both FWHM and Pk were added to the graph to improve readability. However, the arrow for Pk is not visually optimal, as it appears somewhat floating within the graph. Although it is mentioned in the legend, it would be better to find a clearer visual solution to link the "Pk" label to its arrow, similar to the "FWHM" arrow.

We thank the reviewer for this helpful remark. We have adjusted the position of the Pk arrow and its label to improve visual clarity and consistency with the FWHM indication.

Typo:

L173: « autofluoresce » instead autofluorescence

L224 and 229: « signalling » instead signaling

L429: « cells/L » unit instead cells mL^{-1}

L511: « impa » instead impact

Figure 4A: add space between period and (s) on x axis

We thank the reviewer for pointing out these typographical and formatting issues. All have been corrected in the revised manuscript

Reviewer #2b (senior co-reviewer):

After thoroughly reviewing the revised manuscript, I can confirm that all of my previous comments and suggestions have been satisfactorily addressed by the authors. I also fully agree with the observations and recommendations made by my co-reviewer.

Regarding Reviewer #1's comments on the experimental design—specifically the impact of the FR and R ratios, which were previously identified as a weak point in the study—I find that the authors have now addressed this concern convincingly. They provided new data demonstrating the effects of varying FR-to-R ratios, and the results show a behavior similar to that observed with the B/R ratio. This addition strengthens the overall validity of the experimental design.

We thank the reviewer for their careful evaluation and positive feedback. We are pleased that the additional data on FR-to-R ratios satisfactorily address previous concerns and further support the study.

At this stage, I have no further comments or suggestions concerning the manuscript.

However, I remain curious about 2 points:

1- In the continuous light experiment (Fig. 1A), the power spectrum of the ratio still shows a weaker, yet seemingly significant, signal in the green, orange, and red wavelength ranges, with a longer period (in seconds). To me, this suggests that the “wobbling” becomes weaker and occurs at a higher frequency as one moves from the blue toward the red region of the light spectrum (with the exception of FR of course). I am wondering whether the signals recorded between approximately 200 and 400 seconds (depending on the wavelength) correspond to different wobbling frequencies, or whether they might instead result from the Brownian motion of the cells. In the latter case, would one expect such an effect to vary with wavelength?

We thank the reviewer for this insightful observation. In the green, orange, and red experiments, the power spectra exhibit multiple peaks of comparable amplitude, indicating that no single oscillation frequency clearly dominates under these conditions. By contrast, the spectra at 430 nm and 780 nm show a single, well-defined peak with a pronounced amplitude, suggesting that most cells oscillate coherently at a similar frequency under these wavelengths.

While one could hypothesize that the multiple peaks arise from distinct subsets of cells oscillating at different frequencies, we cannot confirm this directly. It also seems unlikely that these features result from Brownian motion, as random displacements would not produce reproducible, well-defined oscillatory signals. Overall, the most parsimonious interpretation is that the oscillatory response is weaker and less coherent in the green, orange, and red ranges compared with the blue and far-red conditions.

2- In line with this question, regarding Figure 4 (“Pulsed light experiments”): does the fact that the authors selected a pulsing frequency of approximately 150 s—which corresponds to the main signal observed under continuous B and FR light—introduce a potential element of circular reasoning? I am inclined to think not, since Figure 4A indeed shows a power spectrum peak around 150 s under pulsed R light. However, I would appreciate confirmation of this interpretation, or at least clarification on how to interpret the weaker signals observed at higher frequencies.

The period of the pulsed light (in both the red and far-red experiments) was determined based on the main oscillation period observed in the collective fluorescence emission of a cell suspension, as shown in Figure 3B. Our aim was to reproduce the oscillatory signal that a cell population naturally generates under continuous illumination.

Therefore, rather than representing a case of circular reasoning, this experimental design allowed us to test whether the cells could synchronize with an external periodic light signal and whether this behavior might be mediated by phytochromes. The fact that cells respond to and synchronize with this signal—even beyond the natural oscillation period of the population (see, for example, Figure 3 in Font-Muñoz et al., 2021)—suggests that light can act as an effective coupling mechanism capable of synchronizing the individual wobbling motions of the cells.

Overall, this study is excellent and highly stimulating, although at times somewhat challenging to follow—even though the authors have made substantial improvements compared with the previous version. The methodology is very specific and complex, and the constraints of the manuscript’s length make it difficult to provide a fully comprehensive description without referring the reader to earlier publications where the methods are explained in greater detail. While this approach is understandable, it does make the reader’s (and reviewer’s) task more demanding, as one must frequently cross-reference multiple papers to fully grasp the experimental procedures—sometimes leaving the impression that one must take certain methodological aspects on trust rather than direct verification.

We thank the reviewer for their encouraging assessment of our study and for recognizing the improvements made in the revised manuscript. We also appreciate their understanding of the complexity of our methodology.

We acknowledge that, given the constraints of manuscript length, full methodological details must be referred to in earlier publications. We hope that the explanations provided here allow readers to follow the key aspects of the experimental and analytical procedures. We are grateful for the reviewer’s thoughtful feedback and their appreciation of the stimulating nature of our work.

That said, at this stage, I do not consider it necessary for the authors to include

additional methodological details in the current version of the manuscript. Perhaps, I could invite them to pay extra attention with the notation throughout the manuscript. For instance, Peak amplitude of the power spectrum is noted "Pk" in figure 1A and $P(\omega)$ in fig4C,D.

We thank the reviewer for this helpful suggestion. We have carefully reviewed the manuscript to ensure consistency in the notation of peak amplitude throughout. Specifically, we have standardized the labeling of the power spectrum peaks, making it uniform across all figures. In addition, we have reorganized Figure S2 in the Supplementary Material to enhance clarity and coherence.

Reviewer #3 (Remarks to the Author):
